# The CXCL13 Index as a Predictive Biomarker for Activity in Clinically Isolated Syndrome

**DOI:** 10.3390/ijms241311050

**Published:** 2023-07-04

**Authors:** Steven C. Pike, Francesca Gilli, Andrew R. Pachner

**Affiliations:** 1Department of Neurology, Geisel School of Medicine at Dartmouth and Dartmouth Hitchcock Medical Center, Lebanon, NH 03756, USA; 2Integrative Neuroscience at Dartmouth, Guarini School of Graduate and Advanced Studies, Hanover, NH 03755, USA; 3Department of Epidemiology, Geisel School of Medicine at Dartmouth, Hanover, NH 03755, USA

**Keywords:** CXCL13, B cell, multiple sclerosis, clinically isolated syndrome, radiologically isolated syndrome biomarker, initial clinical demyelinating event

## Abstract

Multiple sclerosis (MS) is a clinically heterogenous disease. Currently, we cannot identify patients with more active disease who may potentially benefit from earlier interventions. Previous data from our lab identified the CXCL13 index (I_CXCL13_), a measure of intrathecal production of CXCL13, as a potential biomarker to predict future disease activity in MS patients two years after diagnosis. Patients with clinically isolated syndrome (CIS) or radiologically isolated syndrome (RIS) underwent a lumbar puncture and blood draw, and the I_CXCL13_ was determined. They were then followed for at least 5 years for MS activity. Patients with high I_CXCL13_ were more likely to convert to clinically definite MS (82.4%) compared to those with low I_CXCL13_ (10.0%). The data presented below demonstrate that this predictive ability holds true in CIS and RIS patients, and for at least five years compared to our initial two-year follow-up study. These data support the concept that I_CXCL13_ has the potential to be used to guide immunomodulatory therapy in MS.

## 1. Introduction

Multiple sclerosis (MS) is an inflammatory, demyelinating, and neurodegenerative disease of the central nervous system (CNS) of unknown etiology. Its clinical phenotype presents with significant inter- and intraindividual variety, and the course of the disease is difficult to predict. The clinical and pathological heterogeneity of MS makes treatment and overall disease management difficult, which illustrates the need for molecular biomarkers of disease states.

MS activity early in the disease, consisting primarily of clinical attacks and new or enhancing brain magnetic resonance imaging (MRI) lesions, is characterized by the recruitment of leukocytes into the CNS, including populations of T cells and B cells [1,2]. As of the most current McDonald criteria, MS diagnosis relies on the observation of lesions or disease activity disseminated across space and time [3]. Patients with an initial clinical demyelinating event (ICDE) are sometimes classified as having clinically isolated syndrome (CIS) [4], while patients with MRI evidence of demyelinating lesions, but no clinical attacks or findings, are classified as having radiologically isolated syndrome (RIS) [5]. These CIS/RIS patients may convert to a diagnosis of relapsing–remitting MS (RRMS) upon observation of further disease activity; however, 20–30% of CIS patients never progress to a diagnosis of RRMS [6]. Currently, there is no way to distinguish those who will convert to RRMS from CIS/RIS from those who will not [7]. The most current revised McDonald criteria facilitate an early diagnosis of MS in patients who present with CIS [3]. Application of these criteria allows early identification of those patients destined to convert to RRMS, but it is usually accompanied by a higher rate of false positive diagnoses. Some clinicians consider the previous Poser criteria as more clinically relevant, as they rely mainly on clinical evidence for dissemination in space and time [8]. CIS/RIS patients with a second clinical event (relapse) are usually regarded as clinically definite MS (CDMS) according to the Poser criteria. Despite being less sensitive because of their higher specificity, the Poser criteria are a stricter set of diagnostic criteria.

First-line disease-modifying therapies (DMTs), such as interferon-beta and glatiramer acetate, have been shown to delay a diagnosis of RRMS [9], and they are frequently initiated in CIS patients. However, some of these therapies are associated with a high risk of adverse events and would be less beneficial in CIS/RIS patients with a lower risk of converting to RRMS. Thus, healthcare settings may benefit from developing and implementing strategies and tools to improve the early identification of those patients likely to develop or not develop RRMS. Ideally, different subpopulations of patients, which can only currently be identified in retrospect, would be recognized at the time of their first clinical/radiological event consistent with CIS/RIS using predictive biomarkers.

B lymphocytes are critical mediators for neuroinflammation in MS, as demonstrated by observations of intrathecal antibody production in MS patients. This is commonly measured quantitatively, using the IgG index and kappa free light chains, or qualitatively, using oligoclonal bands (OCBs) [10,11,12,13]. Furthermore, B-cell-depleting monoclonal antibodies are potent in decreasing MS activity [14,15,16]. The observations of ectopic lymphoid follicles and B cell aggregates in the CNS of progressive MS patients further emphasize the pathogenic role of B cells [17,18,19]. Lastly, the Epstein–Barr virus (EBV), which has been highly associated with the development of MS [20], directly infects and remains latent in B cell populations. Many have speculated that EBV-infected B cells are involved in the aberrant recruitment of B cells or the production of immunoglobulins in the CNS of MS patients [21]. 

Chemokine (C-X-C motif) ligand 13 (CXCL13), a small (10.3 kDa) chemokine, is a major chemoattractant for B cells and other leukocytes that promote B cell activation, such as follicular helper T cells (TFH) [22]. B cells and follicular helper T cells express the receptor CXCR5 on their surface and utilize the chemokine CXCL13 for trafficking [22,23]; thus, intrathecal production of CXCL13 represents a critical contributory step for the development of relapses in MS and the formation of ectopic lymphoid follicles. In animal models of MS, there is a strong upregulation of CXCL13 within the CNS [24,25,26]. Similarly, increased levels of CXCL13 mRNA and protein have been identified within active and chronic inactive MS lesions [27]. We and others have evaluated the clinical utility and potential value of measuring CXCL13 in the serum or cerebrospinal fluid (CSF) as a biomarker for MS and other neuroinflammatory conditions [28,29,30,31]. In our earlier work, we showed that a normalized ratio of CSF CXCL13 to serum CXCL13, I_CXCL13_, is an excellent biomarker for the prediction of future MS inflammatory activity (attacks and new MRI lesions) in the two years following the diagnostic lumbar puncture (LP) [31]. The purpose of our study is to determine if this biomarker can also be used to predict future disease activity in patients with an ICDE to identify which patients will have highly active disease and which patients may have a more benign form of MS. The data presented below demonstrate that I_CXCL13_ is also predictive for MS activity in CIS and RIS patients for 5 years or more.

## 2. Results

The association between CXCL13 levels and demographic and clinical data is shown in Table 1. CIS and RIS patients were assigned to an I_CXCL13_-high or I_CXCL13_-low group based off previously observed cutoffs (see Section 4). There were no statistical differences in known potential confounders of our analysis, such as age, sex, or follow-up time. The time from CIS/RIS to DMT initiation did not vary between groups. We observed that a higher percentage of patients experienced one or more clinical attacks in the I_CXCL13_-high patient group (*p* = 0.002). In addition, these patients were more likely to be OCB positive (*p* = 0.048), and they had more enhancing lesions identified via MRI (*p* = 0.001). This difference in disease activity is best summarized by our finding that patients in the I_CXCL13_-high patient group were more likely to convert to CDMS, as defined by Poser et al. (*p* < 0.001) [8]. There were nine patients with intermediate CXCL13 levels, and, as expected, their MS activity at follow up was intermediate between the high and low CXCL13 patient cohorts.

With our analysis, we also observed that, on average, patients in the I_CXCL13_-high patient group had more MRIs compared to the I_CXCL13_-low group (*p* = 0.034). We wished to test the hypothesis that the increased risk of identifying lesions was not due to an increase in screening. Thus, we derived the percentage of MRIs in which a new lesion was identified by dividing the total number of MRIs where new lesions were identified by the total number of MRI visits. Even when accounting for the increased screening, we still observed an increased lesion discovery in the I_CXCL13_-high patients (*p* < 0.001). To quantify the predictive value of the I_CXCL13_, we calculated the sensitivity and specificity of our stratification in predicting conversion to CDMS, having one or more new lesions, and having one or more clinical attacks (Table 2).

Lastly, to validate our findings that intrathecal production of CXCL13 results in more recruitment of B cells to the CNS, we looked at the relationship between CSF CXCL13 and the total nucleated cell count (TNC) of the sample. We found a significant positive correlation (*p* = 0.011, R^2^ = 0.13), indicating that patients with elevated intrathecal CXCL13 had more leukocyte infiltration to the CNS.

## 3. Discussion

Clinicians caring for people with MS need better biomarkers to aid them in identifying patients at risk of disease progression and to help in its management. CXCL13, a well-known marker of inflammation in cancer and CNS infections [32,33], has been suggested as a potential prognostic biomarker for MS, as its levels are extremely high in patients with this clinically unpredictable disease [28,29,30]. We have recently examined the predictive value of CXCL13 in determining future disease activity in MS patients, particularly focusing on the intrathecal synthesis of the chemokine [31].

Our initial work on the role of intrathecal production of CXCL13 in inflammatory demyelination was in Theiler’s murine encephalomyelitis virus-induced demyelinating disease (TMEV-IDD), a model of progressive MS [34], where we found that both CXCL13 transcript levels in the spinal cord and CSF CXCL13 protein levels were strongly upregulated [25]. We subsequently probed the CSF of MS patients for cytokines and chemokines and found that CXCL13 concentration was the most consistently elevated protein relative to symptomatic controls (SC), as defined by Teunissen et al. [35], compared to 39 other cytokines and chemokines [29]. Furthermore, in the above study, the CSF CXCL13 concentrations appeared to be highest in those MS patients who subsequently developed MS activity, as defined by MS attacks or new or enhancing brain MRI lesions. To assess the utility of CXCL13 more fully as a predictive biomarker, we studied the intrathecal synthesis of the chemokine in 67 MS patients and 67 SC [31]. In contrast to measuring only CSF CXCL13, quantifying intrathecal synthesis of CXCL13 (I_CXCL13_), as in the use of the IgG index in MS patients, corrects for differences in both serum CXCL13 and blood–CSF barrier integrity between MS patients [36], which can influence the passive transfer of serum CXCL13 into the CSF. In non-inflammatory neurological diseases, such as SC, CXCL13 may be produced in the periphery, but not intrathecally. Accordingly, we found that not only was I_CXCL13_ much higher in MS patients than SC, but it was also highly predictive of future MS activity over the 2 years after the LP [31]. In addition, we have shown that levels of neurofilament light chain protein (NfL) in the CSF are elevated in MS patients with higher I_CXCL13_ [31]. CSF CXCL13 and Q_CXCL13_ (CSF_CXCL13_/serum_CXCL13_) were also predictive, but their positive and negative predictive values were not as high as I_CXCL13_ [31]. To determine whether the predictive ability of I_CXCL13_ lasted more than two years after the LP, we initiated the project described above, which demonstrated that the predictive ability of this biomarker in CIS and RIS lasted at least five years or more.

MS activity, as measured by clinical attacks or new or enhancing brain MRI lesions, is generally accepted to be a consequence of inflammation within the CNS, and it is lessened by treatment with current DMTs, all of which target inflammation. B-cell-depleting therapies (BCDTs), including anti-CD20 monoclonal antibodies, such as ocrelizumab and ofatumumab [16], and Bruton’s tyrosine kinase (BTK) inhibitors, such as tolebrutinib and fenebrutinib [37], decrease MS activity and brain lesions. Given that CXCL13 is a critical chemokine for the recruitment of B cells [22], it is not surprising that intrathecal production of CXCL13, as measured by I_CXCL13_, predicts a more active course in people with MS. Thus, I_CXCL13_ may be very helpful in guiding clinicians in their choice of DMT from the 25 FDA-approved medications, which vary considerably in efficacy and risk.

We patterned our approach to the clinical outcome of CIS similarly to the recent MAGNIMS study of CIS, where the primary outcome was the development of CDMS [38]. CDMS in that study was defined by the following: “the occurrence of a second clinical event attributable to demyelination lasting more than 24 h and after an interval ≥1 month from the first attack, with evidence of two separate lesions,” like the Poser criteria definition [8]. The reason for using CDMS rather than defining MS by the McDonald 2017 criteria is that many of our patients with CIS already fulfilled the McDonald 2017 criteria at their initial evaluation with MRI and OCB findings; thus, the development of McDonald 2017 criteria would not represent new MS activity.

This predictive ability may be especially helpful early on during MS in CIS and RIS patients. Increasing evidence points to improved outcomes in active MS patients with the use of high-efficacy DMTs (HE-DMTs) as early as possible in MS relative to moderate-efficacy DMTs (ME-DMTs) [39,40]. However, HE-DMTs are associated with a higher risk profile, and a recent review article by Filippi et al. [41] concluded that “the identification of prognostic factors can be challenging and current knowledge gaps, including validation of biomarkers and treatment algorithms, may limit their implementation in the clinical setting (i.e., HE-DMTs early in MS).” Thus, clinicians and patients may be loath to accept the risks associated with HE-DMTs very early in the disease, unless the likelihood is high that the patient may have a more aggressive course based on a highly predictive biomarker. CIS and RIS may be considered as representing a “window of opportunity” for preventing disabling long-term outcomes [41,42]. The strong predictive ability of I_CXCL13_ raises the possibility of using this biomarker at this early time point to treat I_CXCL13_-positive patients with HE-DMTs, and using ME-DMTs or a “wait and see” approach for I_CXCL13_-negative patients.

Such a biomarker-guided treatment algorithm would fulfill one of the major goals of biomarker development in MS, i.e., a “personalized medicine” approach to the disease [43]. Such an innovative approach will improve the precision of diagnosis for each patient to capture prognosis and to provide an evidence-based framework for predicting treatment response and personalizing patient monitoring.

This study has some limitations. Firstly, this is a single-center study in which CXCL13 quantification was performed in a single laboratory. More experiments need to be conducted to validate these findings in various environments, such as other centers or laboratories. Secondly, our analysis excluded patients with intermediate CXCL13 indices or CSF concentrations. Our results may not be generalizable to patients with moderately elevated CXCL13 indices. Furthermore, more work needs to be conducted to validate this biomarker in a clinical setting and to understand the pathogenic role of B cells in the different stages of MS.

## 4. Materials and Methods

This single-center study at Dartmouth–Hitchcock Medical Center (DHMC) was approved by the Dartmouth Committee for the Protection of Human Subjects (IRB00000682). Patients with an ICDE, i.e., CIS and RIS patients, underwent a lumbar puncture and blood draw for diagnostic purposes. Written informed consent was then obtained from all study participants for the inclusion of their CSF and serum specimens into the DHMC’s Department of Neurology CSF biobank. Biobanking adhered to the Declaration of Helsinki and was approved by the ethical standards committee at DHMC (STUDY00029241).

CXCL13 quantitation in CSF and serum by Luminex using the BioRad SinglePlex CXCL13 kit (#171BK12MR2; Bio-Rad, Hercules, CA, USA) and BioBank details were as previously described [31]. OCBs and albumin concentrations in the CSF and serum were determined by Mayo Clinic as part of routine medical care. The total nucleated cell count (TNC) of the CSF was determined on site at DHMC as part of routine diagnostic procedures.

Clinical definitions of CIS/RIS were determined according to the 2017 McDonald criteria [3]. The term “CIS” was used for patients with an initial clinical demyelinating event (ICDE) irrespective of the results of the initial workup. The CXCL13 index (I_CXCL13_), as a measure of intrathecal CXCL13 production, which accounts for both blood–CSF barrier integrity and movement of serum CXCL13 into the CSF [31,36], was calculated as [(CXCL13_CSF_/CXCL13_serum_)]/[(albumin_CSF_/albumin_serum_)]. All patients underwent LPs before September 2017 and had more than 5 years of clinical follow up.

The patients were divided into two groups: those with low vs. high indices, as defined below. Because the inclusion of serum with CSF in the DHMC BioBank only began in September 2015, and the CXCL13 index calculation requires serum, the 5-year follow-up requirement in this study resulted in two different groups of patients: one with only CSF CXCL13 concentrations available (n = 29), and the other with the CXCL13 index available (n = 18). For those patients for whom index values could be calculated because of the availability of serum, values above 30 were considered positive or “I_CXCL13_-high”, and values below 20 were negative or “I_CXCL13_-low”, as determined by our previous study [31]; intermediate values were considered indeterminate and were not included. To harmonize the data for patients whose index values could not be calculated because of the lack of availability of serum, they were assigned to the low or high I_CXCL13_ groups by assessing the relationship between I_CXCL13_ and CSF CXCL13 in the patients with complete data. In CSF analyses in the period from September 2017 to September 2022 [31], elevated I_CXCL13_ (>30) values were not observed in patients with CSF values below 5 pg/mL. Similarly, elevated I_CXCL13_ was always seen when CSF values were above 14 pg/mL. Thus, samples with only CSF CXCL13 available with values above 14 pg/mL were considered as having a high index, and those below 5 pg/mL were considered as having a low index.

Non-parametric analyses were used for data analysis, including one-way Fisher exact tests for continuous variables, and Kruskal–Wallis tests were used for categorical data. All statistical analyses were performed using RStudio version 4.2.2 with the tableone package version 0.13.2 [44]. *p*-values < 0.05 were deemed to be statistically significant.

## Figures and Tables

**Table 1 ijms-24-11050-t001:** Demographics of CIS and RIS patients in this study. All patients had a clinical follow up of at least 5 years. One-way Fisher exact test was used to calculate *p*-values (Kruskal–Wallis was used for categorical data). Asterisks denote statistical significance levels: * for *p* < 0.05; ** for *p* < 0.01; *** for *p* < 0.001.

	I_CXCL13_ Low	I_CXCL13_ High	*p*-Value	
**n**	20	17		
**I_CXCL13_ (median [IQR])**	5.99 [4.25, 10.42]	67.07 [47.86, 83.76]	0.003	**
*Serum available* (*%*)	6 (30.0)	7 (41.2)	0.512	
***CSF CXCL13*** (pg/mL) (median [IQR])	2.02 [0.78, 4.77]	17.83 [12.50, 28.55]	<0.001	***
**Age** (mean (SD))	41.89 (12.27)	35.51 (10.51)	0.101	
**Sex** = male (%)	3 (15.0)	3 (17.6)	>0.999	
**Diagnosis** (%)				
*CIS*	16 (80.0)	17 (100.0)		
*RIS*	4 (20.0)	0 (0.0)		
**Follow Up Years** (median [IQR])	6.81 [5.38, 7.20]	6.35 [5.56, 6.99]	0.474	
**Converted to CDMS** (%)	2 (10.0)	14 (82.4)	<0.001	***
**OCB Positive** (%)	13 (65.0)	16 (94.1)	0.048	*
**One or more clinical attacks** (%)	1 (5.0)	9 (52.9)	0.002	**
**Number of MRIs during follow-up** (median [IQR])	3.00 [0.75, 4.25]	4.00 [3.00, 7.00]	0.034	*
**Number of MRIs with new lesions** (%)				
*None*	14 (70.0)	4 (23.5)		
*One*	1 (5.0)	5 (29.4)		
*More than one*	0 (0.0)	8 (47.1)		
*No MRIs Performed*	5 (25.0)	0 (0.0)		
**Number of New or Enhancing Lesions** (mean (SD))	0.07 (0.26)	1.82 (1.78)	0.001	**
**Percent of MRIs with new lesions** (%) (median [IQR])	0.0 [0.0, 0.0]	33.0 [17.0, 50.0]	<0.001	***
**Treatment** (%)				
*Dimethyl fumarate*, *DF*	2 (10.0)	0 (0.0)		
*Fingolimod*, *FIN*	0 (0.0)	2 (11.8)		
*Glatiramer acetate*, *GA*	3 (15.0)	0 (0.0)		
*GA* + *IFN*	0 (0.0)	1 (5.9)		
*Interferon*, *IFN*	6 (30.0)	5 (29.4)		
*Natalizumab*, *NAT*	1 (5.0)	2 (11.8)		
*Ocrelizumab*, *OCR*	0 (0.0)	2 (11.8)		
*Ozanimod*, *OZ*	1 (5.0)	0 (0.0)		

**Table 2 ijms-24-11050-t002:** Sensitivity and specificity of our stratification approach to predict outcome measures in the CIS and RIS patients. TP = number of true positives, FP = number of false positives, TN = number of true negatives, FN = number of false negatives.

Outcome	Sensitivity	Specificity	TP	FP	TN	FN
Converted to CDMS	0.86	0.88	14	3	18	2
One or more new lesions	0.83	0.93	13	4	19	1
One or more clinical attacks	0.70	0.90	9	8	19	1

## Data Availability

Data available from this manuscript can be made available upon request to the corresponding author.

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
