# Peer review of "The CXCL13 Index as a Predictive Biomarker for Activity in Clinically Isolated Syndrome"

_ijms, 2023, doi:10.3390/ijms241311050_

Round 1
Reviewer 1 Report
This is an important follow up study that further supports the use of CSF CXCL13 levels as a predictive biomarker of development of CDMS in patients with CIS. It is scientifically sound, well written and the data are clear. I recommend that it is published in its current form.
The research has addressed the question of whether a bioassay of CXCL13 levels in CSF and serum of patients with clinically isolated syndrome or radiologically isolated syndrome (many of whom will subsequently receive a diagnosis of MS) is predictive of subsequent development of MS.
This research is highly relevant for the treatment of MS as it will potentially allow better informed choice of treatment with appropriate DMTs based around the risk of development of MS following initial diagnosis of CIS/RIS.
I do not consider that the methodology needs to be refined in this instance.
In my opinion the authors conclusions are consistent with the evidence presented.
Author Response
We would like to thank this reviewer for their time in assessing our article
Reviewer 2 Report
The manuscript bring very interesting a clinically helpful results on predictive value of chemokine CXCL13 index in patients with multiple sclerosis. According to the results, patients with high CXCL13 index value more probably convert to definitive clinically manifested MS. It is valuable for clinically isolated syndrome as well as for radiologically isolated syndrome with predictive ability for two or five years after first lumbar punction. Also some consideration about therapy can be expected.
The manuscript is well written and documented. Considering CXCL index as biomarker for MS phenotypes, values of sensitivity, specificity and power test should be completed for “Converted to CDMS, One or more clinical attacks, Number of New or Enhancing Lesions, Percents of MRIs with New Lesions” , Table 1, for clearer idea of readers.
Author Response
We firstly would like to thank this reviewer for their assessment of our report. As per their suggestion, we have included values of sensitivity and specificity for ICXCL13 index and the outcome measures. This data can be found in Table 2 in the newest version of the report.
Reviewer 3 Report
This is an interesting study and tries to answer an unmet need in MS care. Study design and methods are clear.
However, I would like the authors to answer the following points:
Major points:
Considering we use the 2017 version of the McDonald's criteria, authors should thoroughly justify why they chose Posner criteria for CDMS. I would like to see this same analysis performed with the current criteria, as an additional table.
Although the authors excluded patients with intermediate CXCL13 levels, if this is to be deployed in the clinic, one would need to know what intermediate levels mean. I'd like to see the data for conversion to CDMS in the intermediate level patient cohort - is it below the high but higher than the low? This would strengthen the argument for CXCL13 guiding DMT choice.
The authors should specify when DMTs were initiated for patients in both groups.
Minor points:
I would suggest removing the correlation with cell count. The R2 is very low and there are few data points.
Lines 221 - 226 should clarify if the cutoffs were derived from the patients included in this paper or for their whole cohort. This data should be presented in a table.
Furthermore, given the current role of Neurofilament light chain in MS prognosis and monitoring, it would be interesting to have this data for this cohort. I do understand it is outside the scope of the paper and I will not fault the authors for not complying with this request.
Line 34 McDonald and not McDonnell
Lin 62 should also mention kappa free light chains
Lin 104 dividing not diving
Table 1 has asterisks. I suppose they are related to the p-value but clarification in the legend is warranted.
Lines154-156 - sentence needs rewriting
